# Control of water for high-yield and low-cost sustainable electrochemical synthesis of uniform monolayer graphene oxide

Jiaqi Guo[1,2,4], Songfeng Pei [1,2,4] ✉, Kun Huang[1,2], Qing Zhang[1,2], Xizhong Zhou[1,2], Jinmeng Tong[1,2], Zhibo Liu[1,2], Hui-Ming Cheng [1,2,3] & Wencai Ren [1,2] ✉

With the rapid development of graphene industry, low-cost sustainable synthesis of monolayer graphene oxide (GO) has become more and more important for many applications such as water desalination, thermal management, energy storage and functional composites. Compared to the conventional chemical oxidation methods, water electrolytic oxidation of graphite-intercalation-compound (GIC) shows significant advantages in environmental-friendliness, safety and efficiency, but suffers from non-uniform oxidation, typically ~50 wt.% yield with ~50% monolayers. Here, we show that water-induced deintercalation of GIC is responsible for the non-uniform oxidation of the water electrolytic oxidation method. Using in-situ experiments, the control principles of water diffusion governing electro-chemical oxidation and deintercalation of GIC are revealed. Based on these principles, a liquid membrane electrolysis method was developed to precisely control the water diffusion to achieve a dynamic equilibrium between oxidation and deintercalation, enabling industrial sustainable synthesis of uniform monolayer GO with a high yield (~180 wt.%) and a very low cost (~1/7 of Hummers' methods). Moreover, this method allows precise control on the structure of GO and the synthesis of GO by using pure water. This work provides new insights into the role of water in electrochemical reaction of graphite and paves the way for the industrial applications of GO.

Graphene oxide (GO) is a monolayer graphene derivative with many unique properties such as excellent dispersibility in polar solvents, high chemical reactivity, easy assembly into various macrostructures (e.g. fibers, membranes and aerogels), and exceptional compatibility with various materials[1–3]. With the rapid development of graphene industry, low-cost sustainable synthesis of GO has become more and more important for water desalination, energy storage[4–6], functional composites[7,8], thermal management[9,10], and biomedicine[11,12]. For instance, the thermally conductive graphene films made from GO have been widely used in smart phones and 5G communication systems since 2018. Recently, GO has also shown a great potential for applications in smart systems[13–15] and neuromorphic devices[16–18]. Currently, chemical oxidation, represented by the Hummers' method[19], is still the main method for the synthesis of GO. However, chemical oxidation has insurmountable drawbacks including severe environmental pollution, explosive risks, long reaction time, and high cost[20,21]. With the increasingly stringent environmental and safety laws, the production of GO by chemical oxidation is increasingly restricted. Moreover, the

[1]Shenyang National Laboratory for Materials Science, Institute of Metal Research, Chinese Academy of Sciences, 72 Wenhua Road, Shenyang 110016, P. R. China. [2]School of Materials Science and Engineering, University of Science and Technology of China, 72 Wenhua Road, Shenyang 110016, P. R. China. [3]Institute of Technology for Carbon Neutrality, Shenzhen Institutes of Advanced Technology, Chinese Academy of Sciences, 1068 Xueyuan Road, Shenzhen 518055, P. R. China. [4]These authors contributed equally: Jiaqi Guo, Songfeng Pei. ✉e-mail: sfpei@imr.ac.cn; wcren@imr.ac.cn

fine-tuning of the structure and properties of GO product is hard to realize by this method although it is essential for meeting the requirements of different applications.

Electrochemical (EC) exfoliation of graphite has long been used for the synthesis of multilayer graphene but it is extremely hard to synthesize GO[22–24]. In 2018, a water electrolytic oxidation method was developed for the safe, environmentally friendly and ultrafast synthesis of GO[25], where sulfuric acid intercalated graphite compound (SA-GIC) was used for EC oxidation in an aqueous SA solution. Different from graphite, SA-GIC impedes the formation of $O_2$ gases from water electrolysis and ensures sufficient highly active oxygen-containing radicals (·O, ·OH, ·OOH, etc.)[26,27], leading to EC oxidation of graphite. However, this method suffers from non-uniform oxidation. Typically, only ~50% of graphite is converted to GO even at a high voltage of 5 V, with ~50% being monolayers, without purification[25]. Furthermore, the non-uniform oxidation issues of EC oxidation method become more serious when scaling up, in particular, in humid environment, but the reason is not clear.

Here, we found that the deintercalation of SA-GIC caused by water absorption (DIWA) from environments and aqueous electrolyte is responsible for the non-uniform oxidation of graphite. Moreover, both the EC oxidation by water electrolysis (OWE) and DIWA are significantly affected by the diffusion of water from the electrolyte to the interior of electrode. The competition between these two processes determines whether SA-GIC could be uniformly oxidized. Based on these understandings, we developed a liquid membrane electrolysis (LME) method to precisely control the water diffusion to achieve a dynamic equilibrium between DIWA and OWE, which enables industrial sustainable synthesis of uniform GO. The yield of GO can reach ~180 wt.%, with >99% being monolayers, and the cost is only ~1/7 of that of the Hummers' method. Moreover, LME allows precise control on the oxidation degree and lateral size of GO and the synthesis of GO by using pure water.

## Results and discussion
### Dual roles of water in the EC oxidation of GIC
In our previous study, we used $^{18}O$ isotope tracing and radical trapping experiments to reveal that the oxygen-containing radicals (·O, ·OH, ·OOH, etc.) generated from water electrolysis when using stage-I SA-GIC (SA-GIC-I) as electrode are responsible for the EC oxidation of graphite[25]. We further studied the EC reactions of graphite and SA-GIC-I in various non-aqueous electrolytes. Notably, no covalent oxygen functional groups could be detected on graphite in all cases (Supplementary table 1 and Supplementary Figs. 1–4). These results confirm that the involvement of water in the EC reaction is necessary for the oxidation of graphite-based materials (Supplementary note 1).

However, the presence of water in air or aqueous electrolyte can induce significant deintercalation of SA-GIC-I (Fig. 1a–d, Supplementary Fig. 5). Moreover, the higher the water content, the higher deintercalation rate. As shown in Fig. 1e, the SA-GIC-I sample can remain stable for more than 5 days at 0% relative humidity in air; when the humidity increased to 11%, it transformed to stage-II SA-GIC within 150 min; and when the humidity increased to 70% the deintercalation took only 50 min (Supplementary Fig. 6, 7 and Supplementary note 2). The deintercalation of SA-GIC-I is much quicker in aqueous solutions (Fig. 1f). It took less than 5 s in SA solution with a concentration below 60 wt.%. However, as the SA concentration increased, the deintercalation time was prolonged until SA-GIC-I reached a relatively stable state at 90 wt.% SA, where the water molecules are chemically bound with SA.

Besides water, similar deintercalation phenomena were observed in other polar solvents such as methanol, ethanol, propanol, isopropanol, butanol, octanol, ethylene glycol, and ethyl acetate (Supplementary note 3 and Supplementary Table 2). In general, the deintercalation rate decreases with decreasing the polarity of the solvents. For instance, the deintercalation of SA-GIC-I took about 100 s in methanol. However, the intercalation state of SA-GIC-I can be preserved for over 24 h in water-insoluble and chemically inert solvents with low polarity such as paraffin oil (PA), petroleum ether, and carbon tetrachloride ($CCl_4$). Thus, these solvents can act as a barrier between SA-GIC-I and humid air (or the aqueous electrolytes) (Supplementary Fig. 8).

To evaluate the influence of deintercalation, we designed an emulsion electrolysis experiment (Supplementary note 4 and 5) to investigate the EC oxidation of SA-GIC with different intercalation stages (Supplementary Fig. 9), where the SA-GIC contacted with only trace amount of water. The SA-GIC samples were fully immersed in PA first, and then trace amount of water was introduced by adding water-in-oil emulsion drops into PA (Supplementary Fig. 10). When the water-containing colloidal particles diffused to the surface of the SA-GIC electrode, the trace water (each colloidal particle contains ~$10^{-16}$ mol water) was released to the electrode surface (Supplementary Fig. 11). Since the amount of water is far less than the threshold for deintercalation (~10 wt.%), this experimental design avoids the deintercalation caused by excessive water and enables a direct examination of the effects of intercalation stage of SA-GIC on the EC oxidation reaction.

Figure 1g-i show the X-ray photoelectron spectroscopy (XPS) spectra, Raman spectra and UV-Vis spectra of the emulsion oxidation products of SA-GIC with three different intercalation stages. Although all products possess the same kinds of functional groups, the content of functional groups differs significantly. The C/O ratio of the product prepared from SA-GIC-I was ~1.8, same as typical GO, while those from stage-II and stage-III SA-GIC were ~3.6 and ~8.8, respectively (Fig. 1g and Supplementary Fig. 12b). Meanwhile, the products demonstrate a significant decrease in Raman D peak intensity and a redshift of UV-Vis peak around 230 nm with increasing the intercalation stage of SA-GIC (Fig. 1h, i), confirming the decreased oxidation degree and structural disorder[28]. These results suggest that SA-GIC-I is the prerequisite for uniform water electrolysis oxidation and GO can be synthesized using pure water as electrolyte (Supplementary Fig. 12).

In the water electrolytic oxidation process, the DIWA of SA-GIC-I is unavoidable since aqueous solutions were used as electrolytes and the SA-GIC-I was exposed in air before EC oxidation. The lack of stable SA-GIC-I under such environments is the main reason for non-uniform oxidation of the water electrolytic oxidation method, resulting in low yield and low content of monolayer GO, in particular, in humid regions. This also explain why it is difficult to produce highly oxidized GO by traditional EC methods, where graphite is used as electrode and SA-GIC-I could not be obtained in electrolytes containing a large amount of water[29]. Thus, the key to realize uniform EC oxidation is to keep the balance of DIWA and OWE.

### Principles governing DIWA and OWE in SA-GIC
To gain a comprehensive understanding on the DIWA and OWE in SA-GIC-I to find the key parameters influencing the two processes, we performed an in-situ investigation to monitor the morphology and structure changes of the SA-GIC-I anode during EC oxidation (Supplementary note 6). The equipment is shown in Fig. 2a and Supplementary Fig. 13. In the experiments, SA-GIC-I was protected by PA, and the reaction state was manipulated by introducing SA aqueous electrolyte into the reaction system. By controlling the amount of added electrolyte, the contact state of SA-GIC-I anode and electrolyte was changed as shown in Fig. 2b–e.

Figure 2b represents the emulsion electrolysis discussed above. The morphology changes of SA-GIC-I during the reaction is shown in Supplement video S1. Initially, the surface of SA-GIC-I showed uniform color of bright blue. After the addition of 100 μL emulsion (water content ~10 wt.%), spotted yellow areas gradually appeared. In-situ Raman spectra indicate that the blue area is SA-GIC-I (Fig. 2b, region①),

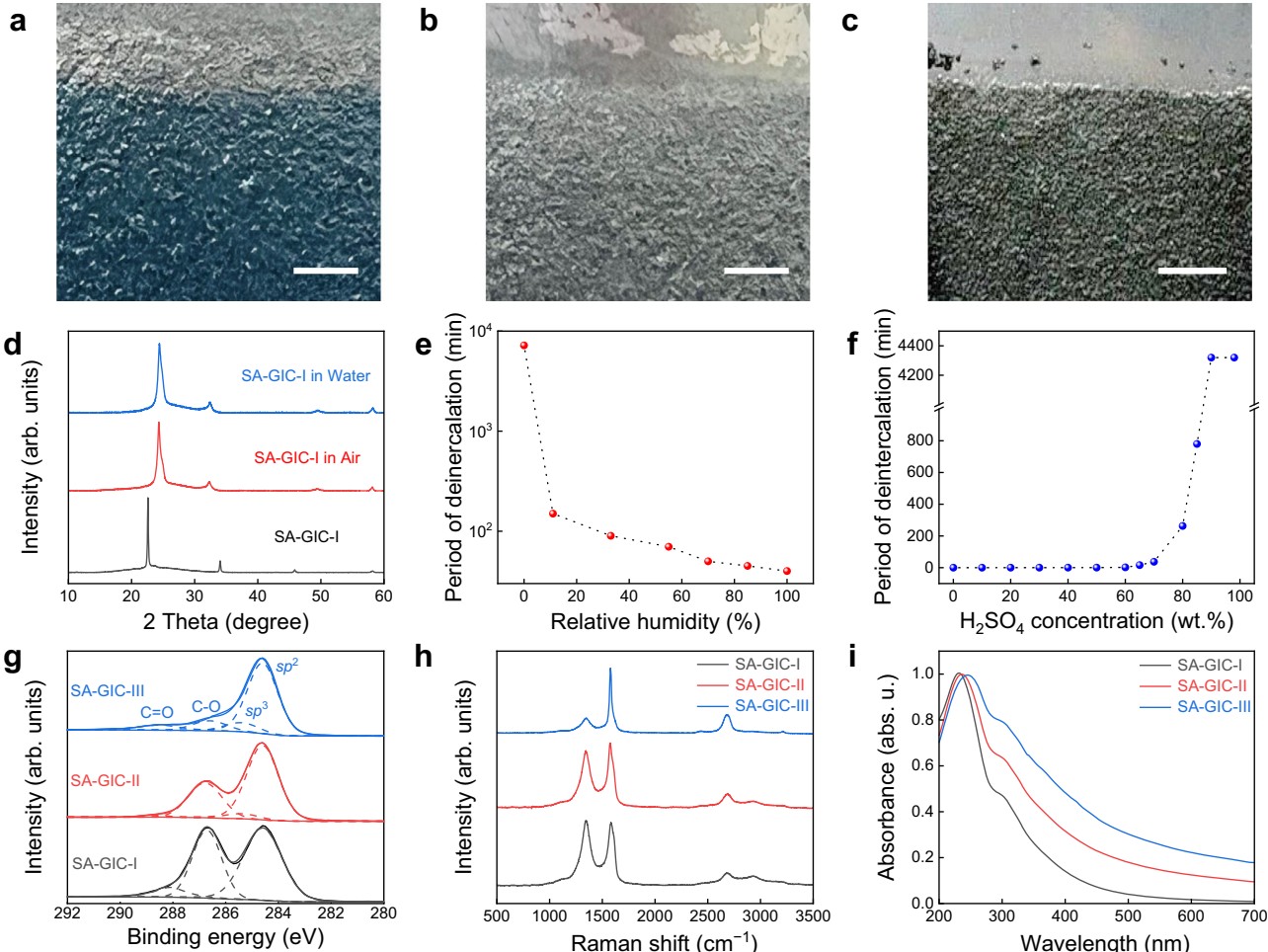

**Fig. 1 | Deintercalation caused by water absorption (DIWA) of stage-I sulfuric acid intercalated graphite compound (SA-GIC-I) and its influence on the properties of graphene oxide (GO) flakes synthesized by electrochemical (EC) oxidation.** Photos of freshly synthesized SA-GIC-I (**a**, blue area), deintercalated SA-GIC by water absorption in air (**b**, light gray area, standing in air with 70% humidity for 50 min at 20 °C), and deintercalated SA-GIC by water absorption in water (**c**, deep gray area, dipping in water for 5 s). **d** The corresponding X-ray diffraction (XRD) patterns of the samples shown in (**a**–**c**). Comparisons of the time required for deintercalation of SA-GIC-I in air with different humidity (**e**) and in SA aqueous solutions with different SA concentrations (**f**). Properties of GO flakes synthesized by emulsion electrolytic oxidation of stage-I, stage-II, and stage-III SA-GIC: (**g**) X-ray photoelectron spectroscopy (XPS) spectra with fitting, where the dashed lines show the deconvolved peaks corresponding to different C binding modes; (**h**) Raman spectra; (**i**) Ultraviolet-visible (UV-Vis) spectra. Scale bars: (**a**), 1 cm; (**b**), 1 cm; (**c**), 1 cm.

and the yellow spots show obvious characteristic peaks of GO (Fig. 2b, region②). After keeping reaction for about 10 s, no further yellow spots appeared until another drop of emulsion was added. The co-presence of SA-GIC-I and GO areas proves that the charged SA-GIC-I can be directly transformed into GO by contacting with trace amount of water without significant deintercalation. Further dropwise adding emulsion to the system, the EC reaction continued until the entire SA-GIC-I anode was completely oxidized into graphite oxide.

Figure 2c, d shows the setups of SA-GIC-I anode partially contacting SA aqueous electrolyte with limited areas, where the content of water is much higher than that in emulsion electrolysis. With these setups, the DIWA and OWE processes can be observed because of the relatively low rate of morphology change. As shown in Fig. 2c and Supplement video S2, when contacting with a 30 wt.% SA solution under 1 V, the blue SA-GIC-I was gradually changed to cyan-white stage-II SA-GIC (Fig. 2c, region②) from the electrolyte/anode interface to the areas far from the interface, which suggests the gradual deintercalation along with the diffusion of water from the electrolyte into the interior of the anode[30]. The serial photos taken at different time points yield a deintercalation rate of ~66 μm s⁻¹ (Supplementary Fig. 14).

The above results suggest that both DIWA and OWE of SA-GIC are controlled by the diffusion of water from the electrolyte to the interior of SA-GIC. Further investigations revealed that increasing the voltage can improve the balance between DIWA and OWE because the intercalated species in SA-GIC are primarily negatively charged $(H_2SO_4)_x(HSO_4^-)_y$[31,32]. As shown in Fig. 2g, the deintercalation rate decreased with the increment of applied voltage within the range of 1.0 –1.9 V, but no oxidation occurred. When the voltage exceeded 2.0 V, oxidized yellow areas appeared, indicating the existence of a minimum threshold voltage at ~2.0 V for EC oxidation, where DIWA was suppressed. Moreover, the oxidation rate increased with increasing the applied voltage. When the applied voltage increased to 2.2–3.2 V, a uniformly oxidized graphite was achieved (Supplementary Fig. 15, Supplementary note 7). However, there is an upper limit of the applied voltage. When the voltage exceeded 3.5 V, the gas generation from water electrolysis became obvious, leading to the detachment of unoxidized particles from the anode.

In addition, the deintercalation rate strongly depends on SA concentration. It exhibits a monotonous downward trend with the increase of SA concentration under a specific voltage (here 2.5 V) as shown in Fig. 2h and Supplementary Fig. 16. This phenomenon can be

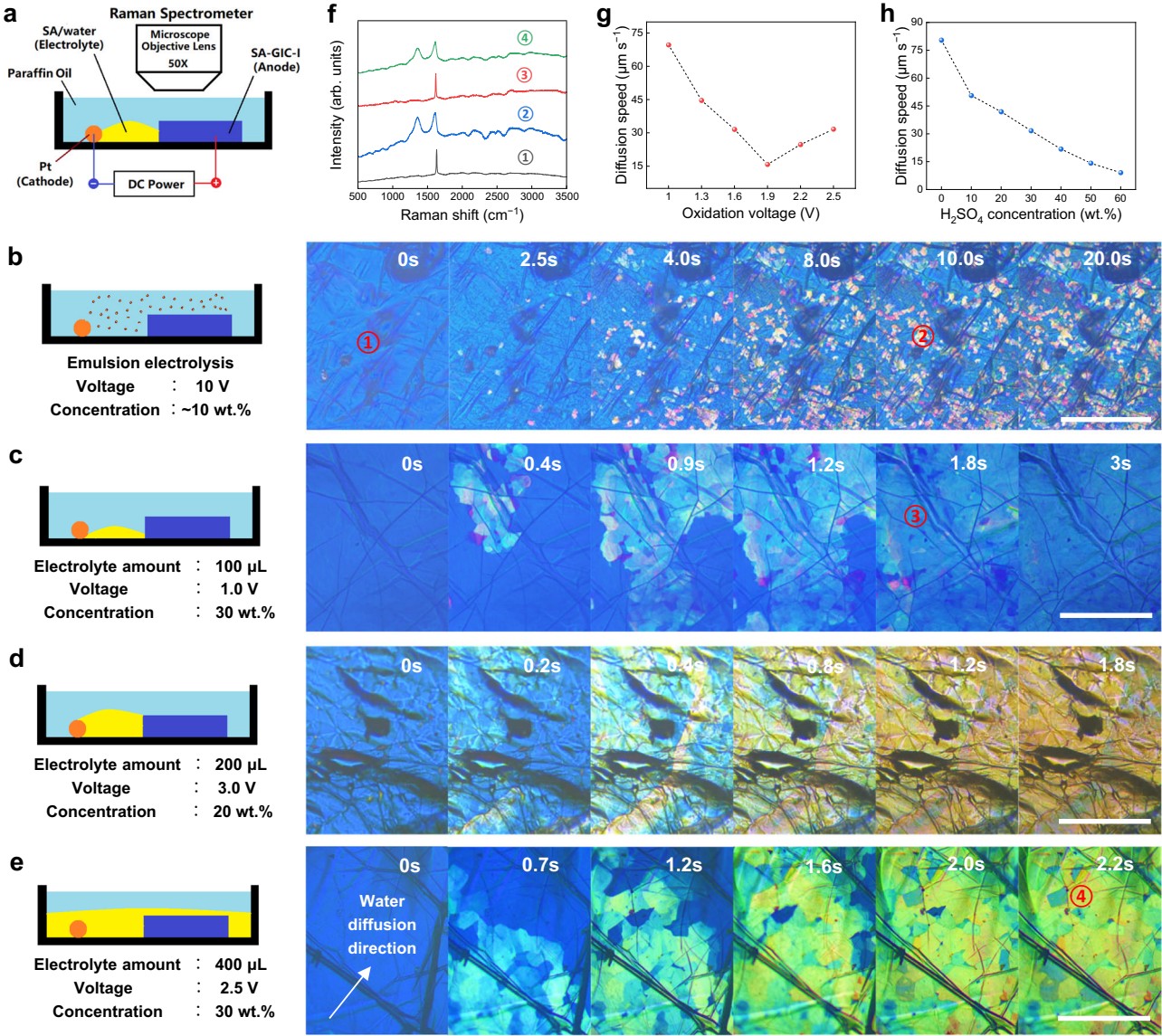

**Fig. 2 | In-situ investigation on the DIWA and oxidation by water electrolysis (OWE) of SA-GIC-I anode during EC oxidation. a** Schematic diagram of the in-situ experimental setup. **b** Setup I: emulsion electrolysis, and the time series photos showing the gradual OWE of SA-GIC-I. **c** Setup II: local contact electrolysis with a small amount of aqueous SA electrolyte, and the time series photos showing the gradual DIWA of SA-GIC-I. **d** Setup III: local contact electrolysis with moderate amount of aqueous SA electrolyte, and the time series photos showing rapid and uniform EC oxidation. **e** Setup IV: electrolysis immersed in excessive aqueous SA electrolyte, and the time series photos showing the rate of DIWA is faster than the rate of OWE, leading to non-uniform oxidation. **f** Raman spectra taken from the labeled areas in (**b–e**). The dependence of water diffusion rate on oxidation voltage (**g**) and SA concentration (**h**). Scale bars: (**b**), 200 μm; (**c**), 100 μm; (**d**), 100 μm; (**e**), 100 μm.

explained from two perspectives: (1) the inhibition of water diffusion due to the decrease of water concentration gradient (Fick's diffusion law) and (2) the decrease of the amount of free water molecules that do not chemically bind with SA molecules. As a result, the deintercalation nearly stopped when the SA concentration exceeded 90 wt.% because most water molecules were bound with SA.

Another important finding on the EC oxidation is that regulating the contact area between electrolyte and SA-GIC-I anode can efficiently control the diffusion rate of water (Supplementary Fig. 17). Increasing the contact area properly not only can keep complete oxidation but also significantly improves the oxidation rate. With setup III, a uniform EC oxidation was achieved rapidly within 1–2 s under 3.0 V using electrolyte with 20 wt.% SA (Fig. 2d and Supplement video S3). The efficiency of this process surpasses that of setup II, with no obvious deintercalation being observed. By further increasing the contact area, SA-GIC-I was fully immersed in the electrolyte, as demonstrated in

setup IV, which is the common state of the existing EC method. In this case, it is difficult to achieve a balance between DIWA and OWE. As shown in Fig. 2e and Supplement video S4, the rate of DIWA is faster than the rate of OWE, leading to non-uniform oxidation of the anode.

## LME method for high-yield synthesis of uniform GO

According to the above results, controlling water diffusion is crucial for achieving a balance between DIWA and OWE, which is the key for the uniform and sufficient oxidation of graphite by water electrolytic oxidation. To this end, four principles should be considered: (1) avoiding the contact between SA-GIC-I anode and humid air, (2) limited interface area between SA-GIC-I anode and aqueous electrolyte, (3) proper voltage applied between anode and cathode, and (4) proper concentration of SA electrolyte.

Based on the above principles, we developed a LME method (Supplementary note 8), where a thin aqueous electrolyte membrane

(density 1–1.2 g mL$^{-1}$, SA concentration ≤50 wt.%) was sealed between a heavy oil layer (density ≥1.2 g mL$^{-1}$, e.g. CCl$_4$, CS$_2$) and a light oil layer (density ≤0.9 g mL$^{-1}$, e.g. PA, petroleum ether) as shown in Fig. 3a, b and Supplementary Fig. 18, 19. This unique sandwich structure not only screened the environmental water but also confined the EC oxidation reaction of SA-GIC-I anode to a limited region that contacted with liquid membrane to avoid the influence of excessive water in the electrolyte. As shown in Fig. 3b, in the liquid membrane area, the blue SA-GIC-I was oxidized into yellow graphite oxide and meanwhile the liquid membrane turned into turbid with a light-yellow color due to the dissolution of a small amount of GO. The reaction proceeded along the anode by continuous injection of heavy oil to lift the liquid film (Supplementary note 9 and Supplementary Fig. 20–22). As shown in Supplementary video S5, the entire SA-GIC-I anode was oxidized from the bottom to the top by gradually raising the liquid membrane within 25 min.

The LME method well solves the non-uniform oxidation issue of the water electrolytic oxidation method (Fig. 3c–l). The as-synthesized GO flakes have similar chemical structure and properties with the GO synthesized by the Hummers' method, showing a C/O ratio of ~1.34 and good dispersion in water (Fig. 3c–f, Supplementary Fig. 23). Importantly, a high GO yield of 181.2 wt.% was achieved, which is comparable to the Hummers' method (Fig. 3l), proving the uniform and sufficient oxidation of the whole graphite anode. Moreover, these GO flakes are over 99% being monolayers with an average lateral size of ~12.4 μm (Fig. 3j–l). Because no metal-containing reactants were used, they are also free of metal impurities such as K and Mn, which are commonly existed in the GO synthesized by the Hummers' method.

Furthermore, the lateral size and oxidation degree of GO flakes can be easily tailored by LME. When maintaining a similar SA concentration, decreasing the voltage led to a notable increase in the lateral size of GO flakes, while their oxidation degree remained nearly unchanged (Fig. 4a, Supplementary note 10). At 2.5 V, we produced large GO flakes with a mean lateral size of ~17.4 μm, with some being larger than 50 μm (Supplementary Figs. 24–26). The oxidation degree of GO can be precisely adjusted by varying the SA concentration in electrolyte (Fig. 4b, Supplementary note 11). The C/O ratio of GO exhibits a V-shaped trend, ranging from ~1.34 to 3.78, as a function of SA concentration (Fig. 4b, Supplementary Table 3 and Fig. 27, 28), but their number of layers and lateral size changed slightly (Supplementary Fig. 29). Moreover, we noticed that reducing the lateral size and increasing the oxidation degree of GO leads to an increase in the absolute value of zeta potential (Fig. 4a, b), suggesting better dispersion of GO sheets in aqueous media, which is beneficial for their processing and applications. This correlation also indicates that zeta potential could be used as a fast and effective tool for in-line monitoring of GO quality during synthesis.

It is worth noting that even pure water can be used to synthesize GO by LME (Fig. 4c–e), which has never been achieved before by the existing synthesis methods of GO. The products are dominantly monolayers (82.4%) with mean lateral size of 9.2 μm, show a relatively low oxidation degree (C/O ratio ~3.78) but similar properties with GO synthesized by the Hummers' method (Fig. 4f–h) and highly EC oxidized GO. As shown in Fig. 4c, d, it exhibits excellent dispersibility in water (absolute value of zeta potential larger than 50 mV) and forms stable liquid crystal phase[33,34].

To evaluate the advantages of LME method for industrial production of GO, we developed an industrial production equipment (Fig. 5a, Supplementary note 12), which enabled continuous mass production of GO in a roll-to-roll manner by using a hundred-meter-long flexible graphite paper (width 1 m, thickness 200 μm) as raw material (Fig. 5b, c, Supplementary Fig. 30). The daily production capacity of GO exceeds 2 kg for a single equipment (Fig. 5d). Similar to the lab-scale LME technology, the GO flakes produced continuously over a period of more than forty days are dominantly monolayers

(~93.8%) with a high yield (~157%) and a high oxidation degree (C/O ratio ~1.40) (Supplementary Fig. 31–33). Note that these industrial-scale values surpass those of lab-scale EC oxidized GO reported (Supplementary Table 4) and most of lab-scale GO synthesized by the Hummers' method (Supplementary Table 5). Furthermore, industrial-scale LME has significant advantages over the industrial-scale Hummers' method in terms of efficiency (~1.3), cost (~1/7), eco-friendliness (~10) and safety (~8) (Fig. 5e, Supplementary note 13). Thus, the LME technology paves the way for the sustainable mass production and application of GO at a low cost.

As an example, we demonstrate the use of industrial-scale EC oxidized GO to produce thermally conductive graphene films, as reported previously[35]. The obtained films show electrical and thermal conductivities (~20000 S cm$^{-1}$ and ~1700 W m$^{-1}$K$^{-1}$) superior to those of the graphene films made from GO synthesized by the Hummers' method (Supplementary Fig. 34, Supplementary note 14), demonstrating their great potential for thermal management of electronics. In addition, an important trend of the research of GO applications relies on artificial intelligence (AI) and machine learning (ML) technologies to optimize the properties of GO-based products such as membranes[36–38]. These approaches offer opportunities to fine-tune synthetic processes and predict properties of GO-based products with high precision. To this end, fine-tuning of the structures and properties of GO flakes is a prerequisite, which needs precise control of the GO synthetic process. Our work about the controlled synthesis of GO by LME, together with the use of AI and ML technologies, would greatly advance the industrial applications of GO in various fields in the future.

## Methods
### Materials
Flexible graphite paper (FGP, JL-AQC-5C, 99.9%) were purchased from Beijing Jinglong Special Carbon Technology Co., Ltd., and fully dried in an oven at 80 °C for 24 h before use. GO sheets synthesized by the Hummers' method (HGO) were purchased from Shandong Leadernano Technology Co., Ltd. Concentrated sulfuric acid (SA, H$_2$SO$_4$, Analytical reagent [AR], 98 wt.%), phosphorus pentoxide (AR, P$_2$O$_5$), methanol (AR, CH$_3$OH), ethanol (AR, C$_2$H$_5$OH), glycerin (AR, C$_3$H$_8$O$_3$), propanol (AR, C$_3$H$_7$OH), butanol (AR, C$_4$H$_9$OH), ethyl acetate(99.5%, C$_4$H$_8$O$_2$), chloroform (99% CHCl$_3$), carbon disulfide(AR, CS$_2$), petroleum ether (AR, C$_5$H$_{12}$O$_2$) and sodium nitrate (AR, NaNO$_3$) were purchased from Sinopharm Chemical Reagent Co., Ltd. Paraffin liquid (C$_{25}$H$_{43}$NO$_3$, density: 0.84–0.86 g cm$^{-3}$), octanol (AR, C$_8$H$_{17}$OH), phosphorus pentoxide (P$_2$O$_5$, 99.99% metals basis), lithium chloride (LiCl, 99.99% metals basis), magnesium chloride (MgCl$_2$, 99.9% metals basis), potassium carbonate (K$_2$CO$_3$, 99.99% metals basis), magnesium nitrate (Mg(NO$_3$)$_2$, 99.9% metals basis), strontium chloride hexahydrate (SrCl$_2$•6H$_2$O, 99.99% metals basis), potassium chloride (AR, KCl, 99.5%), sorbitan trioleate (C$_{60}$H$_{108}$O$_8$, Span-85, 100%,) were purchased from Shanghai Aladdin Biochemical Technology Co., Ltd. Potassium hexafluorophosphate (KPF$_6$, 99%), lithium hexafluorophosphate (LiPF$_6$, 99%) and tetrachloromethane (CCl$_4$, AR) were purchased from Shanghai Macklin Biochemical Technology Co., Ltd. 1-Butyl-3-methylimidazolium tetrafluoroborate (C$_8$H$_{15}$N$_2$BF$_4$, [BMIm][BF$_4$], 99%) were purchased from Beijing InnoChem Science & Technology Co., Ltd. Platinum (Pt, 99.99%) wires with a diameter of 0.5 mm were purchased from ZhongNuo Advanced Material (Beijing) Technology Co., Ltd and used as cathode in EC intercalation and oxidation processes. Lithium perchlorate (LiClO$_4$)/ polycarbonate electrolyte (0.5 mol L$^{-1}$) was purchased from Nanjing Modges Energy Technology Co, Ltd. Perfluoropolyether([CF(CF$_3$)CF$_2$O]$_x$(CF$_2$O)$_y$, 2.0 g cm$^{-3}$) and perfluorooctane(C$_8$F$_{18}$, 98%) were purchased from Shenzhen Jialede Science & Technology Co., Ltd. Deionized water was used as solvent and reagent which has a resistivity greater than 18.2 MΩ cm. All materials were used as received without purification.

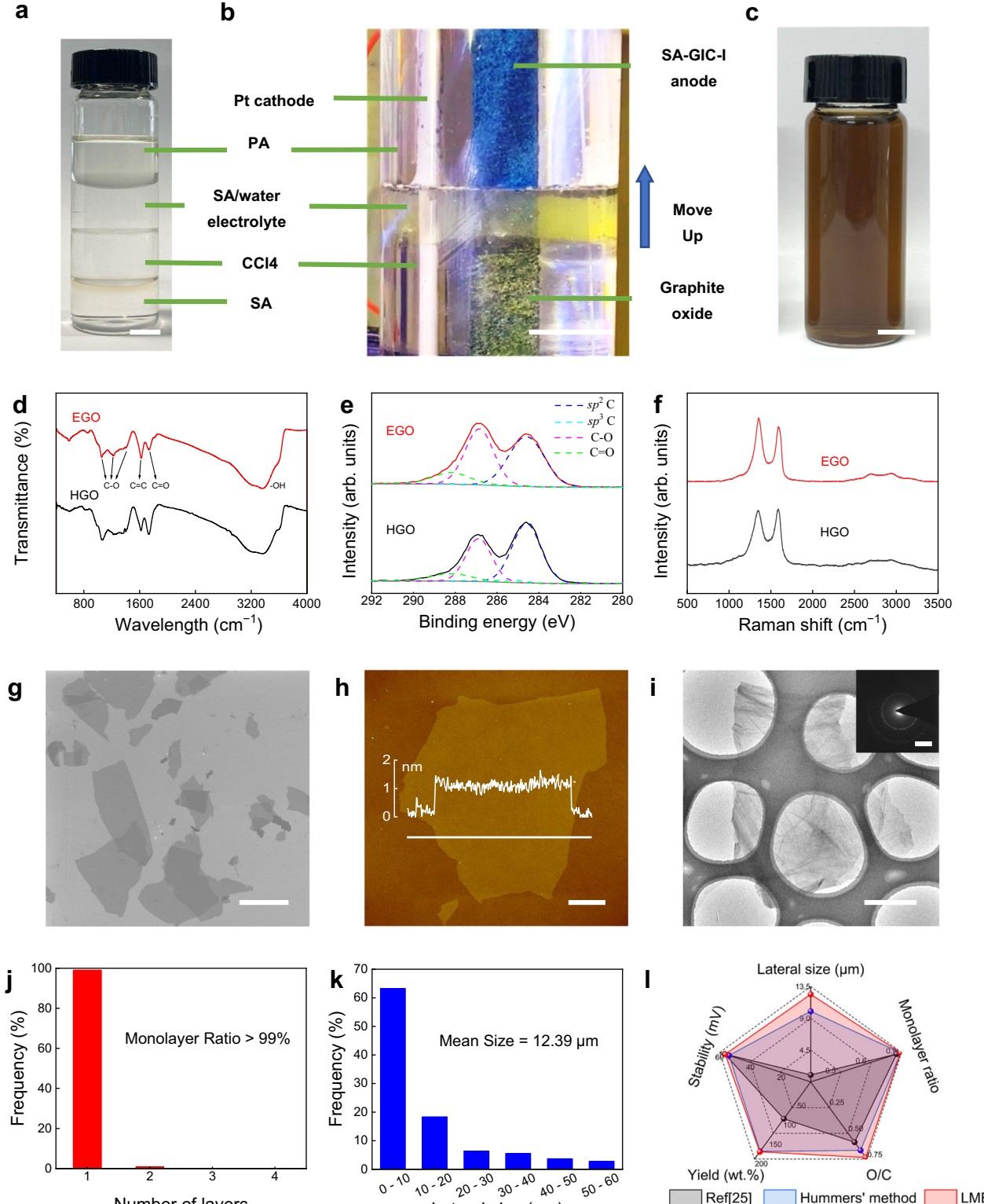

**Fig. 3 | High-yield synthesis of uniform GO by liquid membrane electrolysis (LME). a** Stratification phenomenon of SA, CCl₄, SA/Water, and paraffin oil (PA) based on solubility and density differences (left for 1 month). **b** LME oxidation of SA-GIC-I based on solvent stratification. **c** GO aqueous solution synthesized by LME. Comparisons of fourier transform infrared (FTIR) spectra (**d**), XPS C1s fine spectra with fitting, where the dashed lines show the deconvolved peaks corresponding to different C binding modes (**e**), and Raman spectra (**f**) of GO synthesized by LME (EGO) and the Hummers' method (HGO). Scanning electron microscopy (SEM) image (**g**), atomic force microscopy (AFM) image (**h**), transmission electron microscopy (TEM) image and corresponding selected area electron diffraction (SAED) pattern (inset) (**i**), number of layers distribution (**j**), and lateral size distribution (**k**) of EGO. The inset in (**h**) is the height profile of the EGO flake taken along the white line. **l** Radar plot comparing the existing water electrolytic oxidation method (black), the Hummers' method (blue) and LME method (red), where stability was measured as the absolute value of the zeta potential. Scale bars: (**a**), 1 cm; (**b**), 1 cm; (**c**), 1 cm; (**g**), 10 μm; (**h**), 2 μm; (**i**), 2 μm (inset, 51 nm⁻¹).

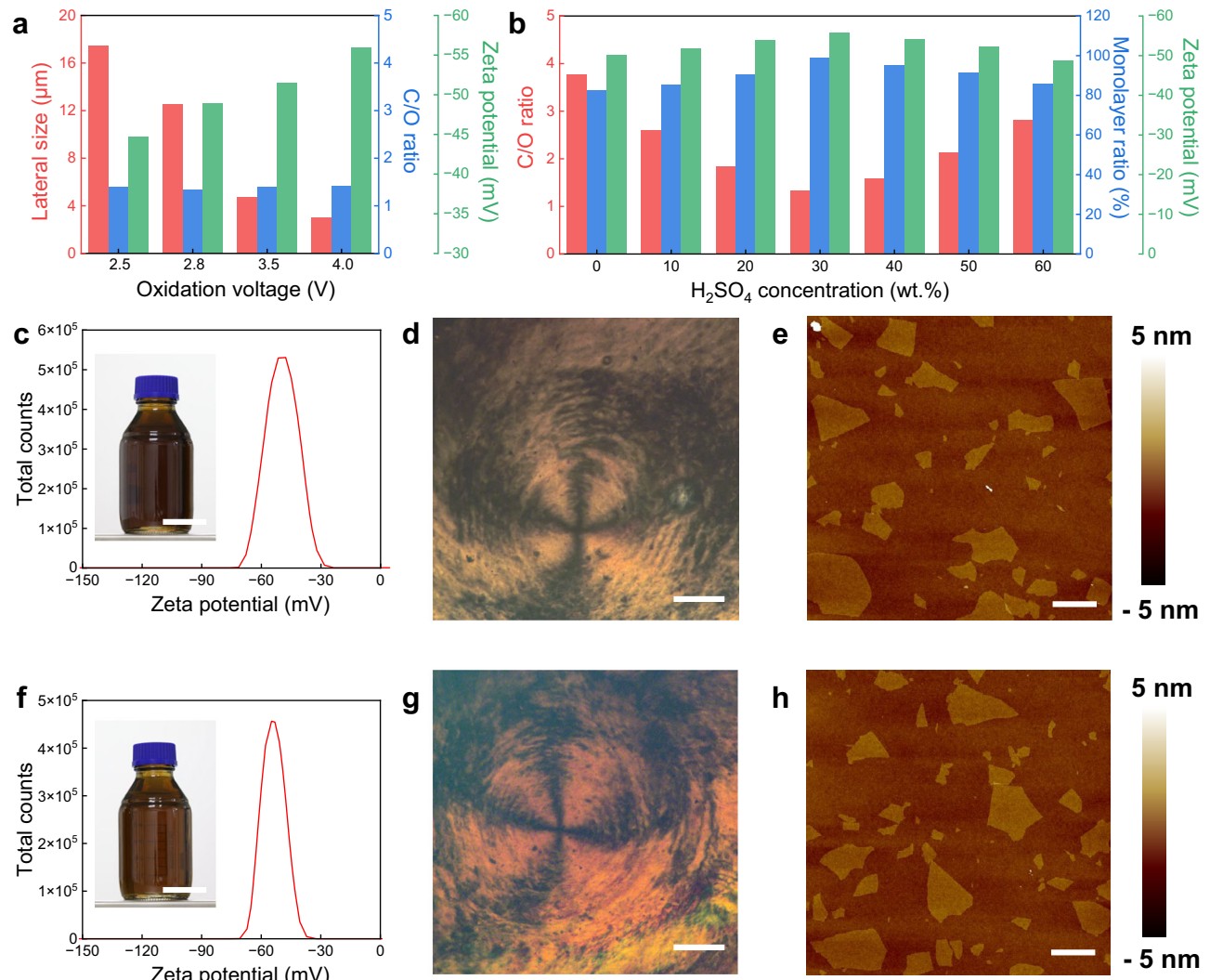

**Fig. 4 | Tailoring the structure and properties of GO by LME. a** The relationship between lateral size, C/O ratio, and zeta potential of GO samples that were synthesized at different oxidation voltages. **b** The relationship between C/O ratio, monolayer ratio and zeta potential of GO sheets that were synthesized at different H₂SO₄ concentrations at an oxidation voltage of 2.8 V. Characterizations of GO synthesized by using pure water at an oxidation voltage of 2.8 V: (**c**), Zeta potential

(inset, the GO dispersion); (**d**), Polarized-light optical microscopy (POM) image; (**e**), AFM image. Characterizations of GO synthesized by using 30 wt.% SA at an oxidation voltage of 2.8 V: (**f**), Zeta potential (inset, the GO dispersion); (**g**), POM image; (**h**), AFM image. Scale bars: (**c**), 5 cm (inset); (**d**), 500 μm; (**e**), 10 μm; (**f**), 5 cm (inset); (**g**), 500 μm; (**h**), 10 μm.

## Characterizations

The microstructure was characterized by scanning electron microscopy (SEM, Verios G4 UC, 10 kV), high-resolution transmission electron microscopy (HR-TEM, FEI Tecnai G2 20), atomic force microscopy (AFM, Bruker, Dimension FastScan with ScanAsystTM, operating in the tapping mode), X-ray diffraction (XRD, Rigaku, SmartLab D/teX Ultra 250 using Cu Kα radiation) and Raman spectroscopy (JY Labram HR 800, 532 nm laser). The chemical compositions were evaluated by combustion elemental analyzer (Elementar, vario MICRO cube), X-ray photoelectron spectroscopy (XPS, ESCALAB 250 using Al Kα radiation source), Fourier transform infrared spectrometer (FTIR, Thermo Scientific Nicolet iS20), thermogravimetric-differential thermal analysis (TG-DTA, Netzsch STA-499C), ultraviolet-visible spectrophotometer (UV-Vis, JASCO V-770) and inductively coupled plasma-optical emission spectrometer (ICP-OES, Agilent 5110). XPS spectra were fitted using the XPS peak 4.1 software in which a Shirley background was assumed. The size of colloidal nanoparticles in water-in-oil colloidal emulsion was measured with Malvern Zetasizer Nano-ZS90. For in-situ microscopic observations, we used an optical microscope (OM, Xi'an Cewei Optoelectronic Technology Co., Ltd) and electrochemical

workstation (CHI 660E). The electrical and thermal conductivities of graphene film were measured by four-point probe system (RTS-9) and LFA467 Hyper Flash (Netzsch, InSb detector), respectively.

### In-situ Raman investigation on DIWA of SA-GIC-I

The SA-GIC-I sample was synthesized by EC intercalation of FGP anodes (slice dimensions: 30 mm length × 10 mm width × 0.2 mm thickness) in 98 wt.% SA for 15 min under a voltage of 1.8 V. Before testing, the sealed reactor was kept at a constant temperature (24 ± 5 °C) for more than 24 hours to obtain a stable humidity environment. Then freshly synthesized SA-GIC-I sample (20 mm × 10 mm) was placed on the dais area in the reactor. After that, the reactor was sealed again and placed on the sample holder of the Raman spectrometer as quickly as possible. The Raman spectra were measured using 532 nm laser with an integration time of 5 s and different time intervals in air with different humidity which was controlled by different saturated salt solutions.

### SA-GIC-I stability test in different solvents

The SA-GIC-I sample was synthesized by EC intercalation of FGP anodes (slice dimensions: 25 mm length × 10 mm width × 0.2 mm

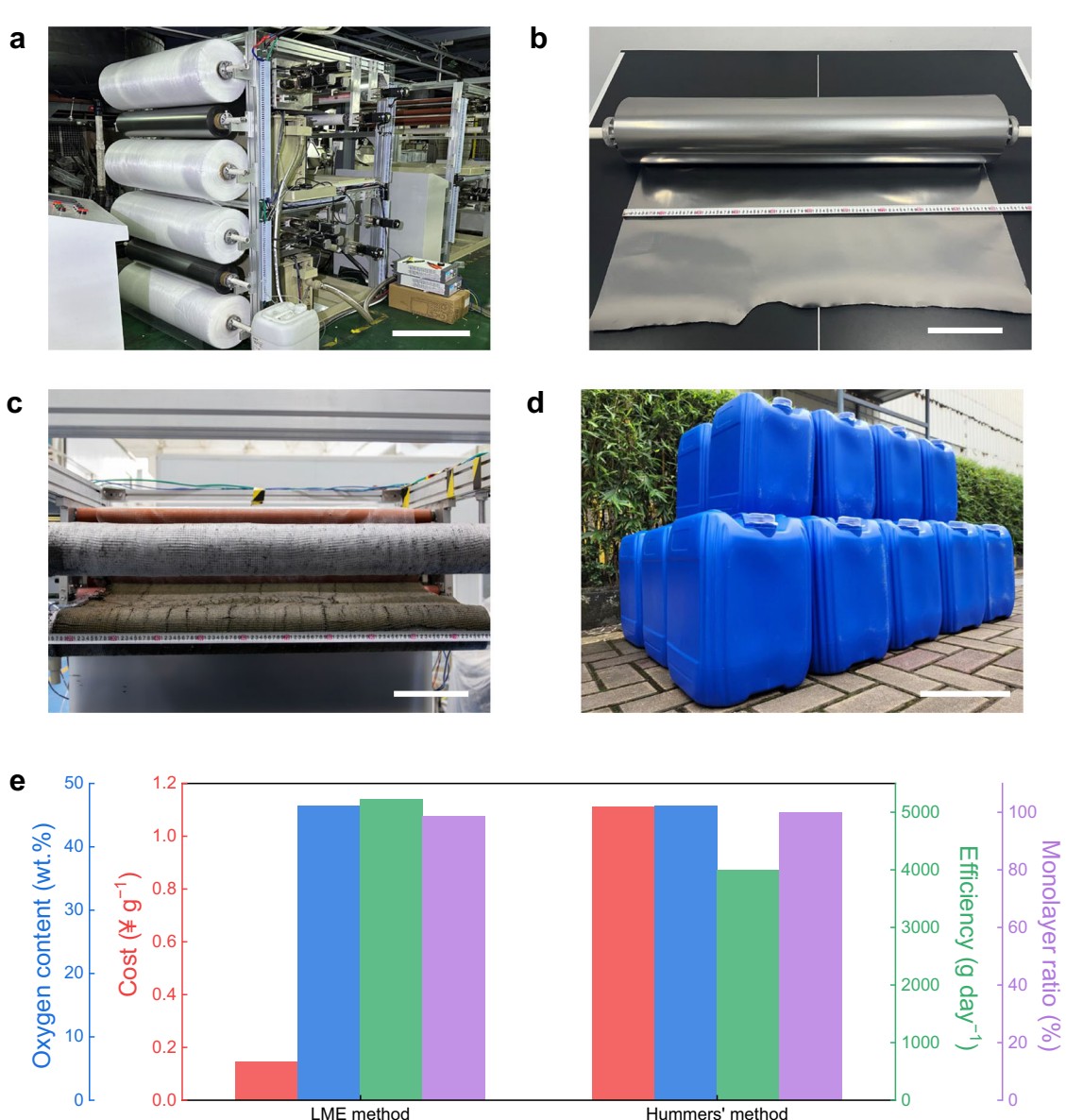

**Fig. 5 | Industrial production of uniform GO by LME. a** Photo of an industrial LME equipment. **b** A graphite paper roll used in the industrial LME equipment. **c** Graphite oxide paper produced by the industrial LME equipment, showing a distinct yellow color. **d** 500 kg GO dispersion (1.2 wt.%) produced by the industrial LME equipment in 3 days. **e** Comparison of the key parameters of LME method and the Hummers' method for industrial production of GO. Scale bars: (**a**), 50 cm; (**b**), 20 cm; (**c**), 20 cm; (**d**), 20 cm.

thickness) in 98 wt.% SA for 15 min under a voltage of 1.8 V. Freshly synthesized SA-GIC-I slices were immersed into different kinds of anhydrous solvents. Since the color change from blue to gray is the typical characteristic of deintercalation of SA-GIC-I, the time needed for the fully fading of blue color of the GIC surface ($T_f$) was used to evaluate the deintercalation rate.

### Emulsion electrolysis
The SA-GIC-I, SA-GIC-II, and SA-GIC-III samples were synthesized by adjusting the intercalation voltages to 1.8 V, 1.4 V, and 1.1 V in 98 wt.% SA, respectively. For the emulsion electrolysis, the distance between the platinum (Pt) cathode and the GIC anode was 5 mm, and the voltage was set to 10 V. An emulsion comprising 10 g of water and 100 g of paraffin oil (PA), emulsified using Span 85 as a surfactant, was prepared, and 200 μL of this emulsion was gradually added to the reactor using a pipette. After 6 hours of reaction, the resulting products were extracted from the reactor and subjected to sonication in water for 20 minutes. The resulting suspensions were centrifuged at 450 g for

three cycles, each lasting 15 minutes. The supernatants were then collected for further characterization.

### In-situ investigation of DIWA and OWE of SA-GIC-I
To observe the DIWA and OWE effects of SA-GIC-I, we performed in-situ observations in a homemade EC reactor. The cathode was a platinum wire with a diameter of 0.5 mm, and the anode was a SA-GIC-I slice with dimensions of 1 cm (length) × 1 cm (width) × 0.1 mm (thickness), and the cathode and anode were simultaneously sealed with PA to prevent the influence of air humidity. The deintercalation and oxidation rates of SA-GIC-I were observed by varying the applied voltage, electrolyte concentration and amount.

### GO synthesis by LME
The synthesis of GO by LME was carried out with the following steps. First, a strip of graphite paper with a density of 1.54 g cm$^{-3}$ and a size of 15 cm (length) × 0.8 cm (width) × 0.2 mm (thickness) was inserted into the homemade LME equipment. Then, 98 wt.% SA was injected into the

reactor, filling it up to 5 mm below the upper port. A small quantity of PA was subsequently added to seal the reactor. Simultaneously, the equipment was powered on with the voltage set to 1.8 V. After approximately 15 minutes of intercalation, all 98 wt.% SA was drained through the lower port and PA was added from the upper port. Once the concentrated SA was expelled, a small quantity of 30 wt.% SA was introduced into the lower port to create a liquid film. This was followed by the injection of $CCl_4$ to elevate the liquid film. Meanwhile, the voltage was adjusted to 2.8 V with a liquid film lifting rate of 1.9 mm min$^{-1}$. After approximately 1 hour, the reaction was complete, and all the $CCl_4$ was drained from the lower port. The obtained graphite oxide strip was immersed in 100 ml deionized water and ultrasonicated for 30 minutes. It was then centrifuged four times at 13000 g for 20 minutes each time. The resulting sample was dried, yielding a mass of 0.515 g. Considering that the length of the graphite paper involved in the reaction was 11.5 cm, its mass was measured to be 0.284 g. Thus, the yield of the sample was calculated to be 181.2%.

### GO synthesis by industrial-scale LME

A roll of flexible graphite paper with a density of 1.54 g cm$^{-3}$ and a size of 85 m (length) × 100 cm (width) × 0.2 mm (thickness) was inserted into the homemade industrial-scale LME equipment. The liquids used in the synthesis process include: perfluoropolyether, 98 wt.% SA, perfluorooctane, 50 wt.% SA, and PA. The graphite paper was converted into graphite oxide paper as it moved through all the above solutions at a speed of 9 cm h$^{-1}$. The applied voltage was 5.0 V in the intercalation region in 98 wt.% SA and 5.5 V in the oxidation region in 50 wt.% SA. Once reaction was complete, the resulting graphite oxide was thoroughly cleaned four times using an industrial-scale filter press operating at 3.5 MPa.

## Data availability

The authors declare that the experimental data supporting the results of this study can be found in the paper and its Supplementary Information file. The detailed data for the study is available from the corresponding author upon request.

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

## Acknowledgements

This work was financially supported by the National Key R&D Program of China (2022YFA1205300 W.R. and 2022YFA1205301 W.R.), the National Natural Science Foundation of China (Nos. 52188101 W.R., 52273240 S.P. and 51872295 S.P.), the Special Projects of the Central Government in Guidance of Local Science and Technology Development (2024010859-JH6/1006 W.R.), the LiaoNing Revitalization Talents Program (No. XLYC2201003 W.R.), and the Guangdong Basic and Applied Basic Research Foundation (2020B0301030002 W.R.).

## Author contributions

W.R. conceived and supervised the project; W.R., J.G. and S.P. designed the experiments; J.G. performed experiments with the help of K.H., Q.Z., X.Z., J.T. and Z.L.; W.R., J.G. and S.P. analyzed the data and wrote the manuscript with input from all authors; H.-M.C. provided advice on the project.

## Competing interests

The authors declare no competing interests.
