## [Transparent Peer Review file · Nature Communications]

Control of water for high-yield and low-cost sustainable electrochemical synthesis of uniform monolayer graphene oxide

Corresponding Author: Professor Wencai Ren

Version 0:

Reviewer comments:

Reviewer #1

(Remarks to the Author)

This work presents a sustainable approach to the mass production of graphene oxide (GO) via electrolysis. The authors claim that the method is economically efficient and demonstrate its potential to significantly improve GO production, addressing a timely and important challenge. The synthetic stages are well characterized, and the work is of high quality. I recommend this paper for publication after addressing the following points:

- 1) GO has exceptional compatibility with various materials, and one of its most important application areas lies in the formation of functional composites for smart systems. Recent advancements, particularly in emerging fields such as neuromorphic devices, further highlight the versatility of GO-based composites. Please include a brief discussion and cite recent publications on GO-based composites for demanding applications like neuromorphic devices to strengthen the manuscript's connection to cutting-edge research.
- 2) The data in Fig. 4 is particularly interesting as it illustrates the complex and intertwined relationships among multiple parameters influencing GO synthesis outcomes. Such convoluted interactions make it challenging to isolate the impact of individual parameters on the overall behavior and results. Among the methods used, z-potential stands out as a fast and effective tool for in-line monitoring of GO quality during synthesis. The observed variations in the z-potential peak under different synthetic conditions provide an excellent opportunity to discuss its utility as a monitoring parameter. Please expand the discussion of Fig. 4 to highlight the role of z-potential in monitoring GO quality. Additionally, the figure caption can be updated to emphasize the importance of these observations.
- 3) Recent studies have demonstrated the effectiveness of AI and machine learning (ML) techniques in optimizing and controlling the properties of GO membranes. These approaches offer opportunities to fine-tune synthetic processes and predict material properties with high precision. Please include a short discussion and cite recent papers on the use of AI and ML for robotization of GO-based membrane technology to demonstrate awareness of emerging trends and potential advancements in this area.

Reviewer #2

(Remarks to the Author)

The authors have presented an interesting and well-conducted study on the dual roles of water in the electrochemical oxidation of graphite intercalation compounds (GIC), culminating in the development of the LME method for producing graphene oxide. The experimental design is robust, and the findings are well-documented, particularly regarding the balance between DIWA and OWE processes and their impact on achieving uniform oxidation.

However, while the work has merit, I believe it may not fully align with the broad scope and high-impact audience of Nature Communications. The research addresses a specific technical challenge in the electrochemical synthesis of graphene oxide and offers a methodologically detailed solution. Yet, its contribution appears to be incremental within the specialized field of graphene oxide synthesis rather than a transformative breakthrough or innovation that would appeal to the multidisciplinary readership of this journal.

To strengthen the impact and positioning of the study, the authors could further emphasize the broader implications of their

method beyond its application to graphene oxide synthesis. For instance, drawing clearer connections to how this approach might revolutionize other electrochemical processes or large-scale material production would broaden its appeal. Additionally, a deeper exploration of how the LME method fundamentally advances the principles of electrochemical oxidation or offers novel scientific insights could elevate the work. Finally possible application can help in improving the manuscript.

Given these considerations, I would recommend to reject the manuscript for this selected journal and that it can be directed to a more specialized journal.

Response to reviewers' comments

Reviewer #1

This work presents a sustainable approach to the mass production of graphene oxide (GO) via electrolysis. The authors claim that the method is economically efficient and demonstrate its potential to significantly improve GO production, addressing a timely and important challenge. The synthetic stages are well characterized, and the work is of high quality. I recommend this paper for publication after addressing the following points:

Response: We thank the reviewer very much for the positive comments and valuable suggestions which have helped us to greatly improve the quality of our work.

1) GO has exceptional compatibility with various materials, and one of its most important application areas lies in the formation of functional composites for smart systems. Recent advancements, particularly in emerging fields such as neuromorphic devices, further highlight the versatility of GO-based composites. Please include a brief discussion and cite recent publications on GO-based composites for demanding applications like neuromorphic devices to strengthen the manuscript's connection to cutting-edge research.

Response: We thank the reviewer very much for the valuable suggestion. We have added a brief discussion and cited a few recently published works about GO-based composites (smart systems and neuromorphic devices) in the introduction section of the revised manuscript.

2) The data in Fig. 4 is particularly interesting as it illustrates the complex and intertwined relationships among multiple parameters influencing GO synthesis outcomes. Such convoluted interactions make it challenging to isolate the impact of individual parameters on the overall behavior and results. Among the methods used, z-potential stands out as a fast and effective tool for in-line monitoring of GO quality during synthesis. The observed variations in the z-potential peak under different synthetic conditions provide an excellent opportunity to discuss its utility as a

monitoring parameter. Please expand the discussion of Fig. 4 to highlight the role of z-potential in monitoring GO quality. Additionally, the figure caption can be updated to emphasize the importance of these observations.

Response: We thank the reviewer very much for the insightful comment and constructive suggestion.

The zeta potential has a close relationship with the dispersion behavior of GO in aqueous media. It is predominantly influenced by the surface charge, which is intimately correlated with the thickness, oxidation degree and lateral size of GO sheets, as well as the pH value of the solution. Thus, it could be used to monitor GO quality to a certain degree. According to the reviewer's suggestion, we have conducted additional testing on the zeta potentials of the GO samples shown in Fig 4a and b in the main text. In all measurements, the GO concentration and the pH value of the solution were maintained at 1 mg mL^{-1} and 6.5, respectively. It can be found that (1) under the similar degree of oxidation, the absolute value of zeta potential showed an downward trend with increasing the lateral size of GO sheets (Fig. R1a); (2) under the similar lateral size, the absolute value of zeta potential increased with increasing the degree of oxidation and monolayer ratio of GO sheets (Fig. R1b).

We have added the above results in the revised manuscript.

Fig. R1 a, The relationship between lateral size, C/O ratio, and zeta potential of GO samples that were synthesized at different oxidation voltages of electrolyte. **b**, The relationship between C/O ratio, monolayer ratio and zeta potential of GO sheets that were synthesized at different H₂SO₄ concentrations.

3) Recent studies have demonstrated the effectiveness of AI and machine learning (ML) techniques in optimizing and controlling the properties of GO membranes. These

approaches offer opportunities to fine-tune synthetic processes and predict material properties with high precision. Please include a short discussion and cite recent papers on the use of AI and ML for robotization of GO-based membrane technology to demonstrate awareness of emerging trends and potential advancements in this area.

Response: We thank the reviewer very much for this valuable and constructive suggestion. Membrane is one of the most important applications of GO, and AI and machine learning (ML) technologies have shown great potential in optimizing and controlling the properties of GO membranes. According to the reviewer's suggestion, we have included a short discussion and cited recent papers on the use of AI and ML for robotization of GO-based membrane technology to demonstrate the emerging trends and potential advancements in this area.

Reviewer #2

The authors have presented an interesting and well-conducted study on the dual roles of water in the electrochemical oxidation of graphite intercalation compounds (GIC), culminating in the development of the LME method for producing graphene oxide. The experimental design is robust, and the findings are well-documented, particularly regarding the balance between DIWA and OWE processes and their impact on achieving uniform oxidation.

However, while the work has merit, I believe it may not fully align with the broad scope and high-impact audience of Nature Communications. The research addresses a specific technical challenge in the electrochemical synthesis of graphene oxide and offers a methodologically detailed solution. Yet, its contribution appears to be incremental within the specialized field of graphene oxide synthesis rather than a transformative breakthrough or innovation that would appeal to the multidisciplinary readership of this journal.

To strengthen the impact and positioning of the study, the authors could further emphasize the broader implications of their method beyond its application to graphene oxide synthesis. For instance, drawing clearer connections to how this approach might revolutionize other electrochemical processes or large-scale material production would broaden its appeal. Additionally, a deeper exploration of how the LME method fundamentally advances the principles of electrochemical oxidation or offers novel scientific insights could elevate the work. Finally possible application can help in improving the manuscript.

Given these considerations, I would recommend to reject the manuscript for this selected journal and that it can be directed to a more specialized journal.

Response: We thank the reviewer very much for the comments, but we cannot agree with his/her concerns about the importance and impact of our work.

GO is one of the most important graphene derivatives with vast number of applications such as water desalination, energy storage, functional composites, thermal management, biomedicine, smart systems, and neuromorphic devices. Our work about the low-cost sustainable synthesis of GO would significantly advance the applications

of GO in various areas and thus it would gain widespread attentions from multidisciplinary audiences. To demonstrate the potential applications and advantages of our electrochemically synthesized GO (EGO), as an example, we fabricated thermally conductive graphene films, where a high thermal conductivity is highly desired for their use in thermal management of electronics such as smart phones and 5G communication systems. For comparison, we also fabricated thermally conductive graphene films under the same conditions by using GO synthesized by the Hummers' method (HGO) as raw materials. As shown in Fig. R2a-c. the EGO-derived graphene films show a smooth surface and highly compact layered structure. The more asymmetrical Raman 2D peak and narrower XRD (002) peak of EGO-derived graphene film suggest its much better crystallinity and ordering than those of HGO-derived graphene film (Fig. R2d, e). As a result, EGO-derived graphene film exhibits much higher electrical and thermal conductivities ($\sim 20000 \text{ S cm}^{-1}$ and $\sim 1700 \text{ W m}^{-1}\text{K}^{-1}$) than those of HGO-derived graphene film (Fig. R2f).

We have added the above data in the revised manuscript.

Fig. R2 The structure and properties of EGO-derived graphene films. a, Photograph of an EGO-derived graphene film. **b,** Cross-sectional scanning electron microscopy (SEM) image. **c,** SEM image of the surface. **d-f,** Comparisons of Raman spectra (**d**), X-ray diffraction patterns (**e**), and electrical and thermal conductivities (**f**) of EGO- and HGO-derived graphene films.